# On Broken Ne(c)ks and Broken DNA: The Role of Human NEKs in the DNA Damage Response

**DOI:** 10.3390/cells10030507

**Published:** 2021-02-27

**Authors:** Isadora Carolina Betim Pavan, Andressa Peres de Oliveira, Pedro Rafael Firmino Dias, Fernanda Luisa Basei, Luidy Kazuo Issayama, Camila de Castro Ferezin, Fernando Riback Silva, Ana Luisa Rodrigues de Oliveira, Lívia Alves dos Reis Moura, Mariana Bonjiorno Martins, Fernando Moreira Simabuco, Jörg Kobarg

**Affiliations:** 1Graduate Program in “Ciências Farmacêuticas”, School of Pharmaceutical Sciences, Faculty of Pharmaceutical Sciences, State University of Campinas (UNICAMP), R. Cândido Portinari 200, Prédio 2, Campinas CEP 13083-871, Brazil; isadora.bpavan@gmail.com (I.C.B.P.); andressa2401@gmail.com (A.P.d.O.); pedrofirminodias@gmail.com (P.R.F.D.); fernandabasei@gmail.com (F.L.B.); kazuo.bio@gmail.com (L.K.I.); fernandoriback@hotmail.com (F.R.S.); aluisa1702@gmail.com (A.L.R.d.O.); l156284@dac.unicamp.br (L.A.d.R.M.); mbonjiorno@gmail.com (M.B.M.); 2Graduate Program in “Biologia Funcional e Molecular”, Department of Biochemistry and Tissue Biology, Institute of Biology, State University of Campinas (UNICAMP), Campinas 13083-857, Brazil; camilaferezin7@gmail.com; 3School of Applied Sciences, State University of Campinas (UNICAMP), Limeira CEP 13484-350, Brazil; simabuco@gmail.com

**Keywords:** DNA damage response, cell cycle, kinase, protein kinase

## Abstract

NIMA-related kinases, or NEKs, are a family of Ser/Thr protein kinases involved in cell cycle and mitosis, centrosome disjunction, primary cilia functions, and DNA damage responses among other biological functional contexts in vertebrate cells. In human cells, there are 11 members, termed NEK1 to 11, and the research has mainly focused on exploring the more predominant roles of NEKs in mitosis regulation and cell cycle. A possible important role of NEKs in DNA damage response (DDR) first emerged for NEK1, but recent studies for most NEKs showed participation in DDR. A detailed analysis of the protein interactions, phosphorylation events, and studies of functional aspects of NEKs from the literature led us to propose a more general role of NEKs in DDR. In this review, we express that NEK1 is an activator of ataxia telangiectasia and Rad3-related (ATR), and its activation results in cell cycle arrest, guaranteeing DNA repair while activating specific repair pathways such as homology repair (HR) and DNA double-strand break (DSB) repair. For NEK2, 6, 8, 9, and 11, we found a role downstream of ATR and ataxia telangiectasia mutated (ATM) that results in cell cycle arrest, but details of possible activated repair pathways are still being investigated. NEK4 shows a connection to the regulation of the nonhomologous end-joining (NHEJ) repair of DNA DSBs, through recruitment of DNA-PK to DNA damage foci. NEK5 interacts with topoisomerase IIβ, and its knockdown results in the accumulation of damaged DNA. NEK7 has a regulatory role in the detection of oxidative damage to telomeric DNA. Finally, NEK10 has recently been shown to phosphorylate p53 at Y327, promoting cell cycle arrest after exposure to DNA damaging agents. In summary, this review highlights important discoveries of the ever-growing involvement of NEK kinases in the DDR pathways. A better understanding of these roles may open new diagnostic possibilities or pharmaceutical interventions regarding the chemo-sensitizing inhibition of NEKs in various forms of cancer and other diseases.

## 1. Introduction

Human NEKs (never in mitosis A (NIMA)-related kinases) are a family of protein kinases composed of 11 members that share structural homology with *Aspergillus nidulans* NIMA proteins [1,2]. NEKs are predominantly related to the cell cycle (mitosis and meiosis), centrosome organization, and primary cilia functions, but also to gametogenesis [3], mRNA splicing [4,5], myogenic differentiation [6,7], inflammasome formation [8], intracellular protein transport [2,9], mitochondria homeostasis [5,10,11,12,13,14,15], and DDR [2,9]. Later studies extended this role to DDR for NEK1 to NEK4, 5, 8, and 10 and then to all other NEKs. There are several classical [16,17] and recent reviews on NEKs functions [2,18] and their role in different diseases [9]. Characteristics of NEKs at the gene and protein levels, such as gene location, number of amino acids, molecular weight, functions, subcellular location, protein domains, and other structural information, are shown in Table 1. In this review, we focus on the emerging family-wide functions of NEKs in DDR.

Genome replication and maintenance are extremely important for cellular functions and the survival of all living organisms. However, during normal cellular metabolism, the DNA is susceptible to chemical modifications by reactive molecules that can cause DNA damage. The sources of spontaneous DNA damage include reactive oxygen species (ROS), deamination, and the inherent susceptibility of DNA to depurination and depyrimidination. In addition, many environmental factors can cause DNA injuries, such as ionizing radiation (IR), ultraviolet (UV) radiation, or chemical agents. These injuries can induce cell death or lead to mutations, which are consequently related to several human disorders, including cancer and age-related diseases. To avoid this situation, cells are equipped with anti-DDR mechanisms and systems capable of repairing DNA damage [19,20].

Based on phosphorylation signaling pathways, the DDR maintains genome integrity through a set of DNA repair mechanisms, damage tolerance processes, and cell-cycle checkpoint pathways. The major functions against DNA damage are DNA repair mechanisms that include: O6-methylguanine-DNA methyltransferase (MGMT) [21], base excision repair (BER) [22], nucleotide excision repair (NER) [23], mismatch repair (MMR) [24], DNA single-strand break repair (SSBR) [25], which relate to the TDP1, APTX, and PNKP proteins and DNA double-strand break repair (DSBR), with two different pathways—the homologous recombination (HR) and non-homologous end-joining (NHEJ) [19].

In eukaryotic cells, cell division undergoes two successive processes. During the S-phase, the DNA is replicated and, during mitosis, the chromosomes are segregated. The gap phases (G1 and G2) take place between S-phase and mitosis. The cell cycle checkpoints control cell cycle events, which include cell size control, chromosome replication and integrity, and the correct mitosis segregation [26].

The DNA damage promotes cell cycle arrest for proper DNA repair. The main proteins related to the checkpoint are ATR, ATM, Chks (checkpoint kinase), CDKs (cyclin-dependent kinases), and p53. Upon DNA damage, these proteins orchestrate the cell cycle arrest through an organized network to properly repair DNA, or to lead the cell to apoptosis [26]. The G1 checkpoint is controlled by ATM-CHK2-p53, while control of the S and G2/M checkpoints are performed by ATR-CHK1-Wee1 [27,28]. The phosphatases Cdc25A and Cdc25C are phosphorylated by CHK2, promoting cell cycle arrest through CDKs [27], which are inactivated after DNA damage until the damage is resolved [26].

The protein kinase mutated (ATM) and Rad3-related (ATR), and DNA-dependent protein kinase (DNA-PK) can recognize DNA damage and start the protein kinase cascade. It is known that double-strand breaks (DSBs) induced by IR are mostly recognized by ATM while UV or stalled replication forks are recognized by ATR [29]. The histone H2AX is phosphorylated by DNA-PK, ATM, and ATR, resulting in the accumulation of γH2AX around the DNA lesion (DSBs) [27,30,31]. The γH2AX induction plays a crucial role in DDR, where the histone H2A variant H2AX is phosphorylated at the serine 139 and recruits DNA repair proteins [29,32,33].

The p53 protein, a tumor suppressor, plays a crucial role in DDR by regulating a large number of genes, related to DNA damage repair, cell cycle arrest, apoptosis, and senescence [34,35]. The CDK inhibitor p21 is transcriptionally activated by p53 and promotes cell cycle arrest and tumor growth suppression [36]. The choice between DNA repair or apoptosis relies on DNA damage type and extent [34,35].

Aside from transcriptional and post-translational mechanisms for DDR, a large part of the DDR is mediated by post-transcriptional mechanisms regulating mRNA processing and metabolism [37,38]. Alternative splicing plays a major role in the control of DDR-related gene expression by affecting both mRNA stability and protein activity. These genes that encode sensing factors, as well as components of the DNA repair, cell-cycle, and apoptotic machinery, often produce splice variants that harbor distinct and sometimes opposite activities [39,40].

Splice variants for CHK1, MDM2 (an important regulator of p53 activity), and CDC25C [41,42,43], among others, have been described in the literature. They are important for the control of cell fate after DNA damage, driving cell cycle arrest for DNA repair or apoptosis.

In addition to their roles in alternative splicing and the expression of DDR genes, a number of splicing factors are recruited at DNA damage sites, indicating their direct participation in the DNA repair process [39]. Among the members of the human NEKs group, the involvement of NEK2 and NEK4 has been observed in the regulation of alternative splicing examples. Most members of the NEK family have demonstrated direct involvement in DDR or interference with partners of paramount importance. We describe below how each member of this family acts in DNA damage and repair. Before presenting each NEK member, will present their main molecular, structural and functional aspects in a table which can be consulted for comparison and overview (Table 1).

**Table 1 cells-10-00507-t001:** Summary of the main molecular features of the members of the NEK family.

NEK Members	Gene Location (Chromosome)	Amino Acids; Molecular Weight	Functions	Subcellular Location	Protein Domains	3D Structure; Method; PDB Entry	Ref.
**NEK1**	4q33	1258 aa, 143 kDa	Primary cilium formation, meiosis I spindle assembly, mitochondrial membrane permeability, cell cycle control, DNA damage response	Cytoplasm, cilia, centrosome, and nucleus upon DNA damage	Catalytic domain, coiled-coil, degradation motif (PEST sequence)	Yes; X-ray; PDB: 4APC	[3,11,36,44,45,46,47,48]
**NEK2**	1q32.3	NEK2A: 445 aa, 48 kDa	Centrosome integrity and separation, cell cycle regulation, primary cilia, splicing	Centrosome, cytoplasm, nucleus	Catalytic domain, coiled-coil, degradation motif (PEST sequence)	Yes; X-ray and electron microscopy; PDB: 2W5H	[4,49,50,51,52,53,54,55,56,57,58,59]
		NEK2B: 384 aa, 44 kDa		Centrosome, cytoplasm	Catalytic domain, coiled-coil		
		NEK2C: 437 aa, 50 kDa		Centrosome, nucleus	Catalytic domain, coiled-coil, degradation motif (PEST sequence)		
**NEK3**	13q14.2	506 aa, 56 kDa	Cell cycle regulation	Cytoplasm	Catalytic domain, degradation motif (PEST sequence)	No	[60]
**NEK4**	3p21.1	NEK4 I1: 841 aa, 94 kDa	Microtubule stabilization, primary cilia stabilization, DNA damage response, splicing	Cilia, basal bodies, nucleus, mitochondria	Catalytic domain	No	[13,61,62,63]
		NEK4 I2: 781 aa, 88 kDa					
		NEK4 I3: 752 aa, 84 kDa					
**NEK5**	13q14.3	708 aa, 81 kDa	Centrosome disjunction, DNA damage response, mitochondrial respiration, mtDNA maintenance	Cytoplasm, centrosome, mitochondria	Catalytic domain, dead-box helicase-like domain, coiled-coils	No	[14,64]
**NEK6**	9q33.3	313 aa, 35 kDa	Mitotic spindle and kinetochore fiber formation, metaphase-anaphase transition, cytokinesis, checkpoint regulation	Cytoplasm, nucleus, mitotic spindle, centrosome, central spindle, midbody	Short unfolded interaction region, catalytic domain	Yes; SAXS	[65,66,67,68,69,70]
**NEK7**	1q31.3	302 aa, 34 kDa	Mitotic spindle formation, centrosome separation, cytokinesis, NLRP3 inflammasome activation, DNA telomeric integrity	Centrosome, spindle midzone, midbody	Short unfolded interaction region, catalytic domain	Yes; X-ray and electron microscopy; PDB: 6S76	[65,71,72,73,74,75,76,77]
**NEK8**	17q11.2	692 aa, 74 kDa	Stability and function of the primary cilium, DNA damage response	Cytoplasm, centrosome, cilia, nucleus, perinuclear compartments	Catalytic domain, Regulator of Chromosome Condensation (RCC1) domain	No	[78,79,80,81]
**NEK9**	14q24.3	979 aa, 120 kDa	Mitotic spindle formation, centrosome separation, replication stress response	Spindle poles, centrosome, cytoplasm, nucleus	Catalytic domain, Regulator of Chromosome Condensation (RCC1) domain, degradation motif (PEST sequence), coiled-coil	No	[82,83,84,85,86,87]
**NEK10**	3p24.1	1172 aa, 133 kDa	DNA damage response, mitochondrial metabolism	Associated with mitochondria	Armadillo repeats, coiled-coil, catalytic domain, degradation motif (PEST sequence)	No	[15,88]
**NEK11**	3q22.1	NEK11 Long (L): 645 aa, 74 kDa	DNA replication and DNA damage response	Cytoplasm	Catalytic domain, coiled-coil, degradation motif (PEST sequence)	No	[89,90]
		NEK11 Short (S): 470 aa, 54 kDa		Nucleus, cytoplasm			
		NEK11C: 482 aa, 56 kDa		Nucleus, cytoplasm			
		NEK11D: 599 aa, 69 kDa		Cytoplasm			

## 2. NEK1

NEK1 is the first ortholog of NIMA kinases described in mammalian cells [36]. Understanding the interaction profile of a kinase is essential to postulate possible investigation pathways. Surpili et al. (2003) performed a yeast two-hybrid (Y2H) screening of a human fetal brain cDNA library and identified NEK1 interactors. In the context of cell cycle regulation and DNA damage repair mechanisms, the authors identified interacting partners, such as 14-3-3 protein, ATRX, MRE11, 53BP1, and PP2A subunit B56 [91]. Later, in 2017, Melo-Hanchuk et al. (2017) performed immunoprecipitation followed by liquid chromatography—tandem mass spectrometry (LC-MS/MS) to study NEK1 interactome after DNA damage induced by cisplatin. The authors revealed interactions with important DNA repair pathways, such as homology recombination, nucleotide excision repair, mismatch repair, and the Fanconi anemia pathway [46] (Figure 1).

Polci et al. (2004) first revealed that NEK1 is involved in early DDR after IR, and plays an important role in repair mechanisms, translocating from the cytoplasm to nuclear foci, where it co-localizes with γH2AX and NFBD1/MDC1 [92]. They extended those findings in 2008 by inducing DNA damage into NEK^−/−^ kat2J mice cells, which failed to arrest cells in G_1_/S or G_2_/M and properly repair DNA after damage induced by IR [45]. In 2011, they demonstrated that NEK1^−/−^ and NEK1^+/−^ mutant cells suffered from nuclei abnormalities and aberrant chromosome segregation, therefore upholding the importance of NEK1 for mitotic control and function in centrosome duplication and stability [93]. In this context, to further explore NEK1 in cell cycle control, Patil et al. (2013) demonstrated that loss of NEK1 leads to severe proliferation defects due to a delay in the S-phase and the interaction with Ku80 [94]. Moreover, Spies et al. (2016) revealed that NEK1 is required for the replication fork stability and homologous recombination repair by phosphorylating Rad54 at serine 572 in late G_2_ and that phosphorylation is required for degradation of stalled replication forks [95] (Figure 1).

In humans, ATM and Rad3-related (ATR) kinases play a major role in DNA damage signaling by activating repair pathways after DSBs, replication stress, and other impairments [96]. Interestingly, a primary study demonstrated that NEK1 activity in DDR and checkpoint control might not depend on ATM and ATR, since inactivation of ATM or ATR did not affect NEK1 expression or translocation to the nucleus after DNA damage. Additionally, neither ATM nor ATR activity was altered in NEK1 deficient cells [97]. In contrast, Liu et al. (2013) demonstrated that NEK1 is required to stabilize the ATR-ATRIP complex for efficient DNA damage signaling, stimulating the autophosphorylation of ATR (T1989) and interacting even before DNA damage induction [44]. However, NEK1 depletion does not show the same effects of ATR depletion since ATR deletion leads to early embryonic lethality [98] and NEK1 depletion to pleiotropic effects, as mentioned before [99]. Thus, the DNA damage signaling functions of NEK1 are more likely to be critical in specific tissues.

A novel pathway suggested by Singh et al. (2020) provides a better understanding of the DDR mechanisms of NEK1. The TLK1 > NEK1 > ATR > CHK1 axis resides on the interaction and activation of NEK1 by TLK1 in threonine 141 (T141) to promote an efficient DDR [12,100,101,102] via cell cycle arrest through CDC25A/CDK1 (Figure 1). Human tousled-like kinases (TLK) are involved in processes of replication, transcription, repair, and chromosome segregation [103]. A protein screen to identify interactors of TLK1B revealed NEK1 as a potential target, which was confirmed by immunoprecipitation and in vitro kinase assays to determine that TLK1 phosphorylates NEK1 at T141. They also found that thioridazine (THD), an inhibitor of TLK1, significantly reduced the activation of NEK1 after DNA damage induction by doxorubicin [100]. These findings led to three other studies exploring the relevance of TLK1 > NEK1 axis for prostate cancer progression, suggesting a novel target for treatment [12,101,102].

In the context of other human diseases, NEK1 was found to be involved in the development of amyotrophic lateral sclerosis (ALS), a neurodegenerative disorder that causes the death of motoneurons (MN) [104]. Subsequent studies also demonstrated that NEK1 loss-of-function mutations conferred susceptibility to ALS [105,106]. Higelin et al. (2018) demonstrated that ALS-associated NEK1 mutations led to dysregulation of DDR machinery and increased cell death to motor neurons, suggesting possible novel therapeutic strategies to reduce DNA damage in neurodegenerative diseases such as ALS [107]. Interestingly, a study demonstrated the interaction between C21orf2 (Chromosome 21 open reading frame 2) and NEK1 related to efficient DNA damage repair [108]. Furthermore, C21orf2 was also found as a risk gene associated with ALS [109], which may indicate a possible novel axis to be explored as a treatment for the disease.

## 3. NEK2

The human NEK2 is the most closely related to NIMA (never in mitosis A), and is one of the most studied proteins in the NEK family [2,49].

DNA damage induces cell arrest in the G1, S, or G2 phase of the cell cycle, depending on the phase in which the damage is sensed, allowing the cells to repair their damaged DNA [110,111]. Based on the fact that NEK2 regulates the separation of the centrosomes during the G2 phase, Fletcher et al. (2004) demonstrated that NEK2 activation and expression are decreased in response to radiation in HeLa cells [59]. Moreover, NEK2-induced centrosome splitting is also inhibited after irradiation in a dose-dependent manner and in different cell lines, as well as in response to other DNA-damaging agents such as etoposide. Thus, NEK2 might act as a downstream target of the DDR pathway and decreased NEK2 activity after irradiation inhibits centrosome separation and contributes to the G2 arrest [59].

ATM, as mentioned, is a central mediator of the cellular response to DNA damage produced by IR, and it is also responsible for activating Protein Phosphatase 1 (PP1), a serine/threonine phosphatase, acting in cell metabolism, transcription, and progression of the cell cycle [112,113]. The PP1α subunit is located in the centrosome and has proven to be a regulator of the NEK2 function, the premature division of the centrosome is triggered both by the overexpression of NEK2 and by the inhibition of PP1 [114], suggesting PP1 as a physiological antagonist of NEK2 [113]. The inhibition of NEK2 activity can be observed in HeLa exposed to IR, and this response proved to be dependent on ATM. the activation of ATM leads to dephosphorylation of the inhibitory residue (T320) of PP1, followed by the increase in its activity and its binding to NEK2. The authors proposed that PP1a maintain NEK2 dephosphorylated, which in turn prevents centrosome splitting [113].

The telomeric repeat binding factor 1 (TRF1) is a double-stranded DNA-binding protein [115] that plays a role in controlling the cell cycle and maintaining telomeres [116,117]. Mitotic aberrations, such as centrosome amplification and chromosome instability caused by NEK2 overexpression, require the TRF1 protein [118]. Misaligned chromosomes, due to overexpression of NEK2 during metaphase, were prevented by depletion of TRF1. Exogenous TRF1, added to cells that overexpressed NEK2, caused the cytokinetic failure, showing that mitotic aberrations caused by NEK2 overexpression are probably dependent on TRF1. NEK2 can directly interact and phosphorylate the TRF1 N-terminal, which contains a D/E-rich and the dimerization domain. It is hypothesized that NEK2 positively regulates the stabilization of the shelterin complex or the TRF1 dimerization. Another possibility is that NEK2 can positively regulate the stability of TRF1 through competition with a ligand that negatively regulates TRF1, Tankyrase 1. Therefore, the association between the overexpression of TRF1 and NEK2 is a mechanism that protects cells against aneuploidy and is related to cancer cell progression [118].

Studies have shown overexpression of NEK2 in several human cancers, acting in malignant transformation, tumor progression, metastasis, and drug resistance [119,120,121]. NEK2 functionally suppresses p53-mediated apoptosis to induce tumorigenesis through p53 phosphorylation in S315 and its reduced stability [122]. In addition, NEK2 depletion impairs drug resistance in multiple myeloma cells through inhibition of the PP1/AKT/NF-κB signaling pathway [50,123].

The role of NEK2 in radioresistance was evaluated in HeLa, where NEK2 depleted cells show a significant increase in the tail comet and yH2AX foci formation, indicating that the NEK2 knockdown accelerates DNA damage [124].

Rad51 is an essential modulator of the HR pathway [125]. The formation of Rad51 foci decreased substantially after exposure to radiation in silenced cells for NEK2, indicating that the loss of NEK2 hinders DNA repair. Finally, the knockdown of NEK2 leads to negative regulation of WNT4/1, both on the mRNA and protein levels. This in turn leads to a down-regulation of beta-catenin and, consequently, to greater radioresistance and oncogenesis progression [124] (Figure 1).

Depending on the extent of the DNA damage, apoptotic pathways can be activated or inhibited to direct cell response for DNA repair or apoptosis, and one of the mechanisms responsible for that is alternative splicing. Genotoxic agents may induce either pro-apoptotic or anti-apoptotic splice variants. For example, cyclophosphamide and UV damage may favor the production of pro-apoptotic caspase 9a and Bcl-xS, while several topoisomerase I and II inhibitors increase the synthesis of anti-apoptotic caspase 2S transcripts [126]. Naro et al. (2014) suggested that NEK2 plays a role in the alternative splicing regulation of several SRSF1 target genes involved in cell viability. Characterization of the subcellular distribution of NEK2 highlighted its co-localization with SRSF1 and SRSF2 in nuclear speckles. Moreover, NEK2 interacts and phosphorylates SRSF1, similarly to SRPK1. The knockdown of NEK2 induced expression of pro-apoptotic Bcl-X, BIN1, and MKNK2 splice variants [37]. The role of NEK2 in alternative splicing may involve additional regulation of cell fate after DNA damage.

## 4. NEK3

NEK3 is one of the less-studied members of the family, and its involvement in DDR is still unknown. However, our group showed evidence of a possible role of NEK3 in DDR [127]. Using the Matchmaker Gold Yeast Two-Hybrid system, a screening for NEK3 interaction partners was performed and 65 clones were obtained, whose cDNAs encode 27 different proteins, functionally involved in sumoylation, ubiquitinylation, regulation of transcription, DNA repair, RNA processing, regulation of cell proliferation, invasiveness, and metastasis [127]. One of the identified interactors is PCNA [127], which is involved in the recruitment of proteins in the DNA replication and damaged DNA repair processes [128]. This study therefore proposes that NEK3 may be involved in DDR through PCNA interaction. However, further studies need to be carried out to elucidate the role of NEK3 in DDR.

## 5. NEK4

In 2012, Nguyen and peers described the interaction between NEK4 and DNA-PK (cs) and its substrates Ku70 and Ku80 [63]. The study was looking for proteins important for replicative senescence in human cells. They found that NEK4 loss of function nearly doubles the cell lifespan. As the suppression of NEK4 did not change telomere length, the observed effects could be attributed to the proliferation rate. Indeed, a reduction of p21 transcription was observed in NEK4 knockdown cells associated with the higher proliferation ratio. As the senescence induced by oncogene H-RasV12 was not affected by NEK4 depletion, they tested if DNA damage could induce cell cycle arrest in these cells and, surprisingly, NEK4 knockdown cells were resistant to DSB agents, such as etoposide and gamma radiation. After etoposide treatment, the fraction of DNA-PK (cs) bound to DNA decreased in NEK4-depleted cells, such as H2AX phosphorylation and p53 activation, despite the fact that DNA damage was found unaltered, nor was there any change in ATM activation [63]. This demonstrated an important role of NEK4 in DDR, mainly linked to DNA- PK recruitment (Figure 2). The mechanism proposed by the authors is that NEK4 acts as a scaffold protein that can maintain the interaction between DNA-PK (cs) and Ku70/Ku80, and can also regulate the distribution of DNA-PK [63].

In 2014, our group characterized a new isoform of human NEK4 (GeneBank: KJ592714) [13]. In screening to identify NEK4 isoforms new interaction partners, we have found Ku70/Ku80/DNA-PK (cs) (PRKDC) and PCNA, as previously reported by Nguyen et al., and several other proteins related to DDR, such as TRIM28 (or KAP-1), HERC2, PALB2, PRPF19, H2AX, MDC1, PML, and MCMs (MCM 3, 4 and 5). Aside from NHEJ-related proteins, various proteins that are classical to other types of DDR were found, such as PARP1, XRCC-6 or CSB, and FEN1, indicating that the role of NEK4 is not exclusive to NHEJ. We have also observed NEK4 and PCNA colocalization after UVC irradiation, as well as colocalization with H2AX after IR exposure. In addition, NEK4 presented PML body localization, a nuclear structure enriched with DDR-related proteins [13]. Nguyen and peers, however, did not exploit the molecular role of NEK4 in the DDR context, but one possibility can be related to DNA-PK (cs) or Ku70/Ku80 phosphorylation. We observed no Ku70 phosphorylation in NEK4 kinase-dead expressing cells compared to control or NEK4 wild type expressing cells [13].

We therefore hypothesize two mechanisms by which NEK4 may participate in the NHEJ pathway via maintenance of the DNA-PK complex: (A) In the NEK4 absence, Ku70/Ku80 does not remain assembled at DSB sites and DNA-PK (cs) cannot bind. As a consequence, H2AX phosphorylation decreases, and so does signaling for DNA repair, culminating in a reduction in p53 activation and cell cycle arrest (Figure 2). (B) Ku70/XRCC6 can be a NEK4 substrate after extensive DNA damage and could activate cell cycle arrest. In NEK4 absence, Ku70 would not be phosphorylated, which would lead to a decrease in DDR signaling and ensuing transcriptional upregulation of p21, followed by cell cycle progress, despite DNA damage.

Fell and colleagues (2016) observed Ku70 phosphorylation at S155 after IR and the S155D phosphomimetic expression led to DDR activation, Aurora B inhibition, and cell cycle arrest. Conversely, the constitutive phosphorylation of Ku70 strongly activated p21 and cell senescence was also observed. The authors proposed that Ku70 phosphorylation at S155 occurs under overwhelming DNA damage to prevent cell cycle progress until the cell completes DNA repair. In case the repair is not possible, the persistent Aurora B inhibition would lead to senescence entry or apoptosis [129].

NEK4 could also directly phosphorylate H2AX, KAP1, and MDC1-promoting signal amplification, or these interactions, could occur in a complex bound to DNA. Surprisingly, our interactome study shows NEK4 interaction with other very early sensors of DNA damage, such as PARP-1, CSB, and ERCC-6 [13], suggesting that NEK4 is closely located at DNA damage sites. PARP-1 is common among all DDR pathways. Based on Nguyen and colleagues’ work and our findings, we propose the NEK4 role in multiple DNA repair pathways, mainly NHEJ through DNA-PK complex, long-patch BER by interacting with FEN1 and PCNA, and TC-NER through its interaction with CSB and PCNA (Figure 2).

When analyzing the NEK4 protein-protein interaction network, we found that several of its interactors play important roles in the mRNA splicing process. Our study also revealed endogenous NEK4 localization to nuclear speckles, regions enriched in splicing factors [13]. Liu and colleagues (2019) demonstrated that DNA-PK (cs) translocates from DNA damage foci to nuclear speckles 30 min after DNA damage [130]. Although additional studies must be performed, NEK4, similar to NEK2, can be a cell fate regulator after DNA damage, regulating the alternative splicing.

## 6. NEK5

Prosser et al. (2015) described the implications of NEK5 knockdown in pericentriolar material (PCM) recruitment, microtubule nucleation, centrosome linker disassembly, and timely centrosome disjunction [64]. NEK5 knockdown showed a fourfold increase of interphase cells, which presented micronuclei and contained 0, 1, or 2 centromeres. An increase in binucleated cells and in the frequency of anaphase cells with lagging chromosomes and unsolved sister chromatids that gave rise to chromosome bridges were also observed. These data suggest that spindle assembly is less efficient in NEK5-depleted cells, which is consistent with the increase in prophase/prometaphase duration. This was the first-ever described evidence that NEK5 could participate in the cell cycle and genomic stability [64].

Considering this information, the role of NEK5 in DDR was further explored, as it is centrosome localized and its altered expression leads to chromosome instability. A recent publication from our research group demonstrated that NEK5 affects DDR mediated by etoposide (Figure 2). NEK5-depleted cells escape the G2/M checkpoint arrest and progress to mitosis, even upon DNA damage induced by etoposide treatment. Moreover, NEK5-depleted cells showed a significant increase in DNA breaks compared to control cells, while NEK5-overexpressed cells attenuated the DNA breaks upon etoposide treatment, demonstrating that NEK5 plays an important role in DDR [131].

Indeed, a yeast two-hybrid screening performed by our research group identified two proteins interacting with NEK5 that are involved with the DDR: topoisomerase IIβ and BCLAF1. Topoisomerase II (TOP2) is an evolutionarily conserved enzyme capable of generating reversible DSBs in DNA, which enables the resolution of problematic DNA topological structures that arise during normal cellular processes, such as transcription, replication, and mitosis [132].

Drugs known as TOP2 poisons, such as etoposide, stop the TOP2 activity and trap it, covalently, to the DNA, leading cells to accumulate DNA breaks and eventually dying; those TOP2 poisons are widely used as anti-cancer drugs [133]. The interaction between NEK5 and Topoisomerase IIβ was confirmed by immunoprecipitation and the dynamic of this interaction was evaluated by proximity ligation assay (PLA). Upon etoposide treatment, the interaction between Topoisomerase IIβ and NEK5 is considerably increased, especially during the early DDR [131].

During DNA replication, topoisomerase II decatenates newly replicated sister chromosomes and aids in relaxing positive supercoils that accumulate ahead of the replication forks. Topoisomerase IIβ is normally present uniformly throughout the cell cycle and it may sustain a catenated state of sister chromosomes to assist chromosome condensation and cohesion without subsequently interfering with segregation [134].

Those data indicate that NEK5 might interact with topoisomerase IIβ to halt the cell cycle upon DNA damage, as this interaction is increased in the early response to damage, leading cells to arrest at the G2/M checkpoint, although this hypothesis should be further explored.

Another NEK5 interactor related to DDR is Bcl-2-associated transcription factor 1 (BCLAF1). BCLAF1 was first identified as a transcriptional repressor that interacts with the anti-apoptotic proteins, Bcl-2 and Bcl-xL, promoting apoptosis when overexpressed. Studies potentially support a role for BCLAF1 in apoptosis through events that control transcription. Despite its role in cell death, it is known that BCLAF1 also participates in other cellular processes, such as T-cell activation, lung development, and RNA metabolism, including DDR [135,136].

BCLAF1 first manifested association with DDR through its interaction with γH2Ax in the IR-induced DSBs model. BCLAF1 promoted apoptosis of irreparable cells through disturbing p21-mediated inhibition of Caspase/cyclin E-dependent, mitochondrial-mediated pathways. BCLAF1 also co-localized with γH2AX foci in nuclei and stabilized the Ku70/DNA-PK (cs) complex, which are DNA damage sensory proteins, facilitating NHEJ-based DSB repair in surviving cells [137]. Later, BCLAF1 manifested its involvement in the selective splicing and export of a large subset of transcripts related to DDR, such as BRCA2, FANCD2, FANCL, and RAD51 [138]. Since NEK5 has been related to both apoptosis and DDR processes, and the roles of BCLAF1 are based on whether cells will correct the DNA damage or undergo senescence or apoptosis, more experiments are needed to better understand the relationship between these two proteins. NEK5 could either activate BCLAF1 to trigger the transcription of DDR genes in an early phase of the DDR, or interact with BCLAF1 to promote cell death via mitochondrial-intrinsic pathways, in which NEK5 had already manifested its role [14].

Finally, NEK5 also proved to interact with Cyclin A2 in breast cancer cells. NEK5 silencing promoted downregulation of Cyclin A2 and Cyclin B1 [139], two of the main mitotic cyclins that are strictly related to mitotic commitment [140].

## 7. NEK6

The knowledge regarding NEK6 involvement in DDR and DNA repair is still sparse. Lee et al. (2008) found that NEK6 kinase activity is negatively regulated in HeLa cells after genotoxic stress [70]. The authors observed phosphorylation of ectopic NEK6 and inhibition of its kinase activity after IR and UV-irradiation, and this phosphorylation was hampered with caffeine treatment 30 min before irradiation. Caffeine is a known inhibitor of the ATM/ATR-Chk1/Chk2 signaling [141] (Figure 3).

In HeLa cells synchronized with nocodazole, when ectopically expressed, immunoprecipitated NEK6 was more phosphorylated and showed higher activity compared to non-synchronized cells [70]. Nocodazole disrupts microtubule dynamics and mitotic spindle function by binding to β−tubulin and inducing cell cycle arrest at G2/M phase [142], and NEK6 transcription and kinase activity is upregulated during the M phase [65]. However, after IR and UV-irradiation, the phosphorylation of ectopic immunoprecipitated NEK6 was increased in both synchronized and non-synchronized cells, but the kinase activity diminished when compared to synchronized non-irradiated cells [70]. NEK6 phosphorylation by the DDR checkpoint proteins Chk1 and Chk2 were also described in an in vitro assay [70].

The necessity of NEK6 kinase activity for mitosis is known since kinase-dead NEK6 expression blocks chromosome segregation at the metaphase-anaphase transition before increasing apoptosis in more than half of the cells [69]. Conversely, flow cytometry analysis of HeLa cells overexpressing NEK6 demonstrates a loss of cell cycle arrest in G2/M after IR-irradiation [70]. NEK6 overexpression induces a reduction of the G2/M population with an increase of G1 and sub-G1 population after IR compared to irradiated wild-type cells, while overexpressing only NEK6 does not change cell cycle distribution [70]. The same is observed with etoposide treatment [143].

NEK6 expression promotes cell proliferation by regulation of cyclin B transcription levels mediated by Cdc2 [144] Additionally, NEK6 expression counteracts cell cycle arrest, ROS production, and the reduction of cyclin B and Cdc2 induced by topoisomerase inhibitors, doxorubicin, and camptothecin treatments, and by p53 expression [145,146].

Therefore, this evidence indicates that inhibition of NEK6 kinase activity during the G2/M phase after genotoxic stress is necessary for proper cell cycle arrest, which is at least partly explained by the phosphorylation of NEK6 by the caffeine-sensitive ATM/ATR-Chk1/Chk2 signaling pathway (Figure 3).

Several miRNAs are predicted and some are confirmed to target NEK6, such as miR-23a [147], which is induced by DNA damage [148]. MiR-23a is induced by berberine treatment, a potent genotoxin that induces the accumulation of DSBs [149,150], in a p53-dependent manner in the HCC HepG2 cell, repressing NEK6 and inhibiting p53 repression induced by NEK6 [147] (Figure 3).

Although NEK6 can be classified as a high confidence hub kinase and the interactors are related to several signaling pathways, including DDR and repair [2,151], to date, no study has identified molecular signaling implications related to DDR after the NEK6 kinase activity inhibition by genotoxic stress. However, NEK6 was found to colocalize at the centrosome with RAD26L, CDC42, and TRIP4, which demonstrated to be NEK6 substrates, as shown by in vitro kinase assays using NEK6 WT and NEK6 S206A (inactive mutant) [151]. In the same study, in a pull-down assay using NEK6 WT and NEK6 Δ1-44 (mutant lacking N-terminal regulatory domain), the authors showed that the NEK6 N-terminal is important for RAD26L, CDC42, and TRIP4 interaction, but not the C-terminal kinase domain [151].

NEK6 phosphorylation by Chk1 and Chk2 is described between amino acids 1–80 [70] since NEK6 S37 was the only residue found to be phosphorylated in vivo in this region [83]. It is therefore likely that NEK6 kinase activity inhibition mediated by Chk1 and Chk2 occurs via the N-terminal regulatory domain [70]; this is the same region that is important for RAD26L, CDC42, and TRIP4 interaction [151]. Thus, it is reasonable to speculate some functional regulation induced by DDR through NEK6 phosphorylation by Chk1 and Chk2, impacting cell cycle, cytoskeleton organization, DNA repair, and NF-κB signaling. It is also dependent on the interaction with CDC42, RAD26L, and TRIP4 at the N-terminal regulatory domain [151].

Zuo et al. (2015) found a NEK6 kinase activity-independent role in the SMAD4 transcriptional function [152]. The authors reported that NEK6 kinase-dead overexpression increased the nuclear localization and transcriptional function of SMAD4 using CAGA-reporter when compared to control cells. In contrast, the overexpression of NEK6 blocked nuclear localization and transcriptional function of SMAD4 in cells treated, or not, with TGFβ when compared to control cells, inhibiting cell cycle arrest induced by TGFβ in a kinase activity-dependent manner [152]. TGFβ is activated after IR-induced DNA damage, and TGFβ treatment increases XPA and XPF interaction and nuclear localization with ERCC1, requiring SMAD4 signaling [153,154,155].

Therefore, NEK6 kinase activity-independent roles observed by Zuo et al. (2015) [152] may have some relationship with the inhibition of NEK6 kinase activity after DNA damage [70]. This might also be related to the regulation of the NEK6 interaction with several proteins, such as SMAD4, RAD26L, CDC42, and TRIP4, as described above.

Cisplatin resistance is associated with an increase of NER and HR and with a reduction in MMR [156]. De Donato et al. (2015) found that NEK6 overexpression decreased cisplatin sensibility in A2780 cells, a human ovarian carcinoma cell line [157]. Another study by De Donato et al. (2018) found a compound named compound 8, that binds to NEK6 and NEK1 kinase domains and inhibits their kinase activities, increasing cisplatin sensibility in PEO1 cells, another ovarian cancer cell line [158].

Jeon et al. (2010) [159] identified that the ectopic expression of NEK6 in JB6 Cl41 cells phosphorylates S727 of STAT3 (Signal transducer and activator of transcription 3), increasing transcriptional activation activity [159]. The transcription factor STAT3 is a member of the STAT signaling family and mediates cytokines and growth factor signaling, promoting transcription of genes related to proliferation, invasion, angiogenesis, and anti-apoptosis [156,159,160,161]. STAT3 also modulates ATM-Chk2 and ATR-Chk1 pathways by MDC1 transcriptional regulation [162], and it contributes to cisplatin resistance [163].

## 8. NEK7

The stability of telomeres is very important to protect against diseases such as cancer and prevent premature aging [164]. Cancer cells have altered oxidative metabolism, promoting telomeric DNA damage [165]. Telomeric repeat binding factor 1 (TRF1) is an essential component of the shelterin complex and acts to protect the integrity of telomeres after oxidative damage to DNA [166].

One study showed that NEK7 regulates the integrity of telomeres [77] (Figure 3). It is recruited and accumulated at telomeres in situations of oxidative DNA damage, protecting telomeres against this type of damage. Moreover, γ-H2AX and association of tumor suppressor p53 binding protein 1 (53BP1) to telomeres, after DNA damage, was diminished in NEK7-deficient cells when compared to control cells. NEK7-deficient cells also presented telomere aberrations. Mechanistically, NEK7 can bind and phosphorylate TRF1 on serine 114, which limits its interaction with F-box only protein 4 (Fbx4) and prevents its degradation via the proteasome. Activation of the ATM pathway is required for NEK7 recruitment and its role in maintaining telomeric integrity. Thus, NEK7 regulates the stability of TRF1, thereby protecting the DNA telomeres from oxidative DNA damage [77].

A mass spectrometry analysis identified that RAD50 co-immunoprecipitated with NEK7 in HEK293 cells [167]. RAD50 acts in the repair of DSB [168] and also in the telomeres’ integrity [169]. Therefore, NEK7 may have several functions in response to DNA damage and the integrity of telomeres. Nevertheless, more studies need to be carried out to fully understand the role of this kinase in this context.

## 9. NEK8

Studies have shown that the maintenance of the genome requires NEK8. By employing comet assay, NEK8 knockdown in HeLa cells shows evidence of DSBs. H2AX phosphorylation was also induced by the absence of NEK8 in these cells, especially after aphidicolin treatment, an inhibitor of the B-family DNA polymerases [170]. NEK8^−/−^ MEFs (mouse embryonic fibroblasts) show hypersensitivity to this treatment, presenting a decline in cell viability [171].

NEK8 prevents DSB accumulation by suppressing cyclin A-associated CDK activity, and the interaction of NEK8 with ATR, ATRIP (ATR-interacting protein), and CHK1 proteins are stronger in the presence of aphidicolin compared to control [81] (Figure 4). NEK8 proved to be important for the progression of replication forks, since the low expression of NEK8 in these cells is associated with deficient S-phase progression, therefore indicating its relationship with DNA replication [81].

During Homologous Repair and Recombination, RECA, homolog to RAD51, is the major protein that performs homology search and DNA strand invasion. Additionally, HR is associated with the replication fork support, since RAD51 plays a role in the replication fork protection [19,172] and promotes the connection between the invading DNA and homologous duplex DNA template [173].

NEK8 appears to play a role in HR [174]. Abeyta et al. (2017) demonstrated the regulation of RAD51 focus formation by NEK8 upon DNA damage, using NEK8-depleted human osteosarcoma cells (U2OS) and NEK8^−/−^ MEFs. The lack of NEK8 promotes inhibition of RAD51 foci formation after treatment with MMC (mitomycin C, a DNA cross-linking agent,), IR, and HU (hydroxyurea, an inhibitor of ribonucleotide reductase) [174]. Employing a specific siRNA against NEK8 in U2OS cells, the authors also showed that these cells presented a mild decrease in the RAD51 expression, demonstrating the role of NEK8 in RAD51 foci formation, regardless of the damage type or cell type [92]. To further assess the effect of NEK8 depletion in MEFs after DNA damage, the cells were treated with HU, MMC, PARP inhibitor (AZD2281), ATR inhibitor (VE-821), etoposide, or microtubule inhibitor (paclitaxel). NEK8-depleted cells saw a decrease in cell survival after treatment with HU, VE-821, MMC, and AZD2281 [174]. NEK8^−/−^ cells presented a decrease in HR efficiency, thus showing that NEK8 plays a role in the response to stalled replication forks, mediated by RAD51, and in the maintenance of genomic stability [174]. All these data imply the involvement of NEK8 in DDR.

## 10. NEK9

To date, few studies have observed the association of NEK9 with DDR. NEK9 was identified as a possible ATM/ATR substrate [175]. A proteomic approach has shown that NEK9 is a putative interactor of DDR and DNA replication proteins, such as RFC3, RRM1, MCM5 CHK1, Ku70, and Ku80 [176].

A study employed a siRNA library to identify genes involved in the replication stress response, which sensitizes to gemcitabine treatment [87]. Gemcitabine is a nucleoside analog used as a chemotherapeutic agent and its incorporation into replicating DNA causes a halt in replication since new nucleosides can no longer be added, leading to the arrest of tumor cell growth and induction of apoptosis [177]. Smith et al. (2014) showed that NEK9 knockdown caused hypersensitivity to gemcitabine in a non-cell-type-specific manner [87]. The knockdown of NEK9 also caused hypersensitivity to other agents involved in replication stress, such as MMC, HU, and camptothecin, a topoisomerase I inhibitor. Moreover, NEK9 may be considered a component of the replication stress response (RSR), since its knockdown causes a spontaneous accumulation of γH2AX and RPA70 foci and a decrease in the recovery from replication stress. Upon replicative stress caused by HU, gemcitabine, and MMC treatments, NEK9 expression usually increases. Finally, NEK9 also interacts in a complex with CHK1 and its knockdown decreased CHK1 autophosphorylation (S296), and its kinase activity in response to replication stress. Thus, NEK9 and other NEKs, such as NEK1, NEK6, NEK8, and NEK11, are involved in RSR and DDR activities [87]. The inhibition of NEK9, concomitant to conventional chemotherapy treatments may be an advantageous strategy since the reduction of its expression sensitizes cells treated with DNA damaging agents.

A previous phylogenetical analysis of all NEKs at large and, specifically, of the vertebrate NEKs had shown the clustering of human Neks in subfamilies [178]. Most interestingly, a distinct subfamily cluster is formed by human NEKs 8,9,1,3, and 5. It is remarkable that three of these NEKs: NEK1, 8, and 9, are all somehow functionally involved with the ATR/ATM axis (Figure 1 and Figure 5).

## 11. NEK10

Moniz and colleagues (2011) showed the role of NEK10 in DDR after irradiation with UV light [88]. UV light usually generates two photoproducts, CPDs (cyclobutane pyrimidine dimer) and 6-4 PPs (6-4 pyrimidine-pyrimidone), which are, in general, repaired by the NER system [20,179,180,181] (Figure 5).

The activation of ERK1/2 (Extracellular Signal-Regulated Kinases) is dependent on MEK1/2 (Dual specificity mitogen-activated protein kinase 1/2) activity, which is in turn activated by the kinase RAF-1 (RAF proto-oncogene serine/threonine-protein kinase) [182]. Moniz et al. (2011) observed an increase of ERK1/2 phosphorylation in HEK293T cells with NEK10 overexpression after UV irradiation, which was dependent on MEK activity [88]. Since NEK10-depletion impairs the activation of ERK1/2 after UV irradiation, NEK10 was implied in mediating ERK1/2 activation. Further investigations showed that NEK10, RAF-1, and MEK1 form a ternary complex. NEK10 also plays a role in the maintenance of the G2/M checkpoint upon irradiation with UV light [88].

Recently, by employing mass spectrometry interactomics, our group found possible NEK10 interactors related to DDR, such as SMC3 (structural maintenance of chromosomes protein 3), UBC (SUMO-conjugating enzyme UBC9), ATRX (transcriptional regulator ATRX), PRKDC (DNA-dependent protein kinase catalytic subunit), and SUMO1 (small ubiquitin-related modifier 1) [15].

Haider and coworkers (2020) showed the importance of NEK10 in p53 phosphorylation [183], which is a tumor suppressor that regulates cell cycle arrest, senescence, apoptosis, autophagy, metabolic reprogramming [35,184], and downregulates genes needed for DNA repair [185]. NEK10 phosphorylated p53 at Y327, increasing the levels of p21 in a NEK10-dependent manner [183]. The knockout of NEK10 by CRISPR promoted greater proliferation, colony formation, and DNA replication. The increase in G1/S phase arrest was also related to NEK10-dependent growth suppression. In the presence of genotoxic stress caused by cisplatin, the lack of NEK10 impaired the induction of p21 and, upon IR, the levels of p21 increased in control cells compared to NEK10 knocked-down cells [183]. These data show the importance of the role of NEK10 in the maintenance of DDR upon genotoxic stress, promoted through UV light, cisplatin, or IR. 

## 12. NEK11

During the cell cycle, the NEK11 subcellular localization is altered and this implies the different roles of NEK11 in the interphase and mitotic phase, in addition to its role in DDR, participating in DNA replication in response to genotoxic stress [89] (Figure 5). NEK11 activity is shown to increase two-fold after treatment with inhibitors of DNA replication (aphidicolin, thymidine, and hydroxyurea) compared to asynchronous cells. After the treatment with DNA damaging agents, such as etoposide, adriamycin/doxorubicin, and cisplatin, the activation of NEK11 was observed, implying its role in DDR [89]. 

The treatment with caffeine decreased NEK11 activation by aphidicolin and other genotoxic agents, suggesting the involvement of NEK11 in the ATM/ATR checkpoint, in addition to a possible role of NEK11L in the phase S checkpoint [89].

The cell division cycle 25 (CDC25) family members are phosphatases responsible for activating cyclin-dependent kinase (CDK) complexes, key regulators of cell cycle progression. CDC25 is involved in cell cycle and checkpoint control [186]. The degradation of the CDC25A is mediated by CHK1, controlling normal cell cycle progression [187]. In mammalian cells, three isoforms are found: CDC25A, CDC25B, and CDC25C [186]. As shown by Melixitian and colleagues, degradation of CDC25A was regulated by NEK11 [177]. They employed an shRNA library to screen for genes that lead to abnormal cell progression into mitosis after IR. In addition to the classical genes responsible for the G2/M checkpoint, they also found NEK11.

The role of NEK11 in DDR of CRC (colorectal cancer) cells was evaluated by Sabir et al. (2015) [90]. Using flow cytometry, they show that NEK11-depleted cells had a decrease in the G2/M phase after IR, suggesting that it may be partially p53-dependent. NEK11 has also been related to the prevention of apoptosis and cell survival [90].

Normal expression of NEK11 leads to increased G2/M arrest, which results in DNA integrity control. However, NEK11 depletion in HCT116 cells prevented G2/M arrest after irinotecan treatment [90], a topoisomerase I inhibitor [188]. This suggests the role of NEK11 in the G2/M checkpoint. In this same study, these cells showed p53-dependent apoptosis, both in the presence and absence of DNA damage, indicating that the loss of NEK11 leads to the induction of an exacerbated p53-dependent apoptosis after IR exposure [96]. Jointly, these data show the importance of NEK11 kinase in DDR.

## 13. Conclusions

NEKs are known for their role in the cell cycle, mitosis, centrosome disjunction, and primary cilia, but not for their participation in DDR. We reviewed the literature and presented important information associating the activity of NEKs in the context of DNA damage, induced by IR or chemotherapeutic agents, for example. NEKs are associated with several known DDR pathways, such as ATM/ATR, CHK1, CDKs, p53/p21, and RAD51. By regulating the cell cycle, NEKs appear to be an important link between DNA damage and cell cycle arrest, guaranteeing a proper response for cells to repair the damage. Considering recent advances in synthetic lethetic as a strategy for cancer treatment, the role of NEKs in DDR shines a light on these kinases. Therefore, NEKs may not only be pivotal for the maintenance of cell homeostasis but may also represent novel opportunities for therapeutic interventions of NEK-related diseases such as cancer.

## Figures and Tables

**Figure 1 cells-10-00507-f001:**
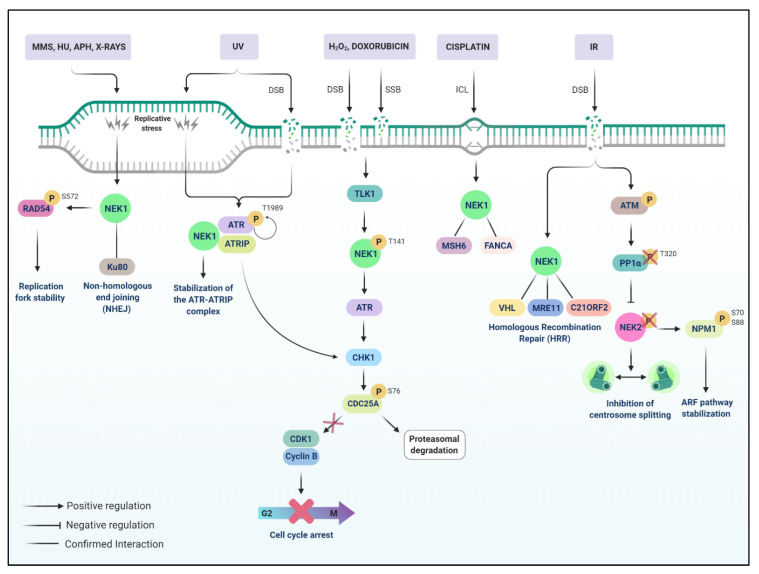
NEK1 and NEK2 play key roles in the DDR pathway. NEK1 is required for replication fork stability by phosphorylating Rad54 at S572. Moreover, the TLK1 > NEK1 > ATR > CHK1 axis plays a major role in DDR, involving cell cycle arrest through CDC25A phosphorylation and CDK1 inhibition. TLK1 phosphorylates NEK1 at T141 and NEK1 interacts with ATR, promoting the stabilization of ATR-ATRIP complex and autophosphorylation of ATR at T1989. NEK2 regulates centrosome disjunction since ATM activates the PP1 phosphatase, which in turn dephosphorylates NEK2, leading to the inhibition of centrosome splitting. The PP1a phosphatase activation also counteracts the NEK2-dependent phosphorylation of NPM/B23, which stabilizes the ARF pathway. NEK2 also upregulates the Wnt1/β-catenin pathway.

**Figure 2 cells-10-00507-f002:**
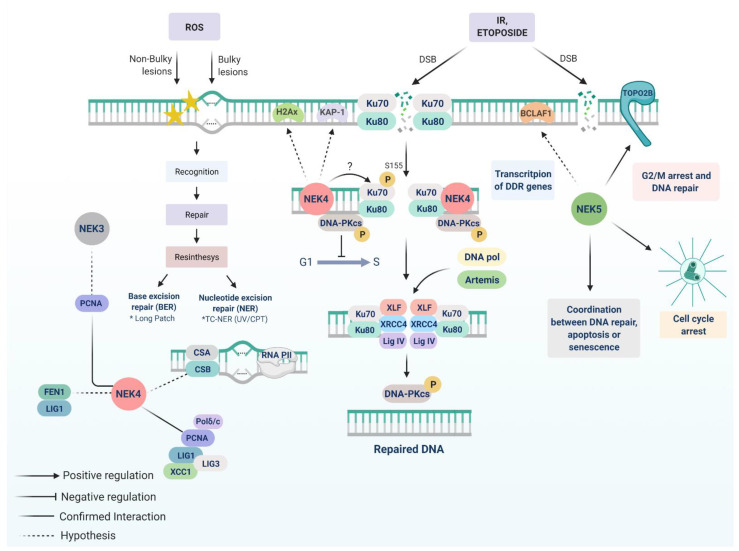
NEK3, NEK4, and NEK5 roles in DDR. The interaction of NEK3 and PCNA is a possible link between NEK3 and DDR. NEK4 interacts with the DNA-PK complex and is important for efficient DNA-PKcs recruitment to DNA damage foci, as well as the activation of pathways essential to induce cell cycle arrest after DSBs with IR or etoposide. We propose that NEK4 acts very early in DNA break recognition, phosphorylating Ku70 (possibly at S155 residue) and stabilizing the DNA-PK complex. Also, NEK4 possibly interacts with other NHEJ-related proteins, such as H2AX, and KAP-1, and PCNA, and FEN1. The interaction between NEK5 and topoisomerase IIβ might be related to cell cycle halting due to DNA damage, as this interaction increases during the first stage of DDR. NEK5-depleted cells overcome the G2/M checkpoint upon DNA damage, indicating that NEK5 is required for an appropriate DDR.

**Figure 3 cells-10-00507-f003:**
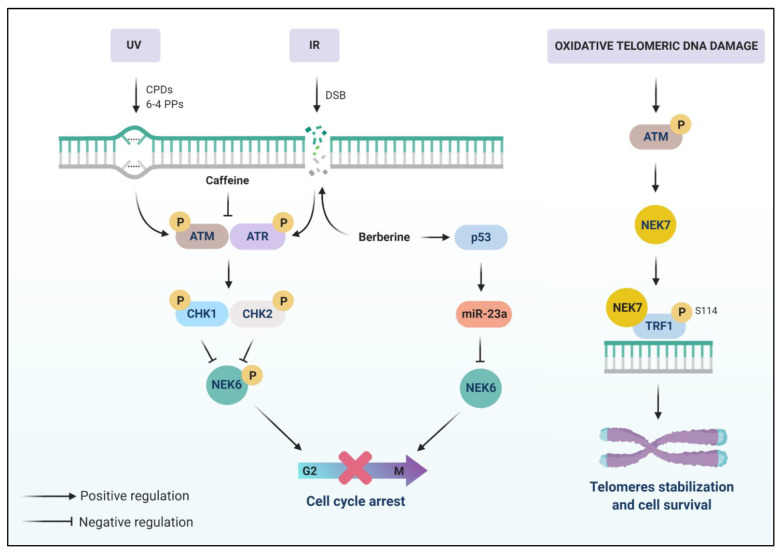
The possible involvement of NEK6 and NEK7 in DDR. NEK6 kinase activity is inhibited by ATM/ATR-Chk1/Chk2 caffeine-sensitive signaling pathway after DSBs induced by IR and UV treatment. NEK6 kinase activity inhibition by DDR is necessary for G2/M arrest. NEK6 is repressed by miR-23a, which is induced by berberine, a genotoxin that increases the accumulation of DSBs, in a p53-dependent manner. NEK7 is related to a protective function of telomeres in response to oxidative DNA damage. ATM activation mediates the function of NEK7 in telomeres. NEK7 is recruited to the telomeres and binds and phosphorylates TRF1 in S114, preventing its degradation. The integrity of telomeres is linked to the stability of TRF1.

**Figure 4 cells-10-00507-f004:**
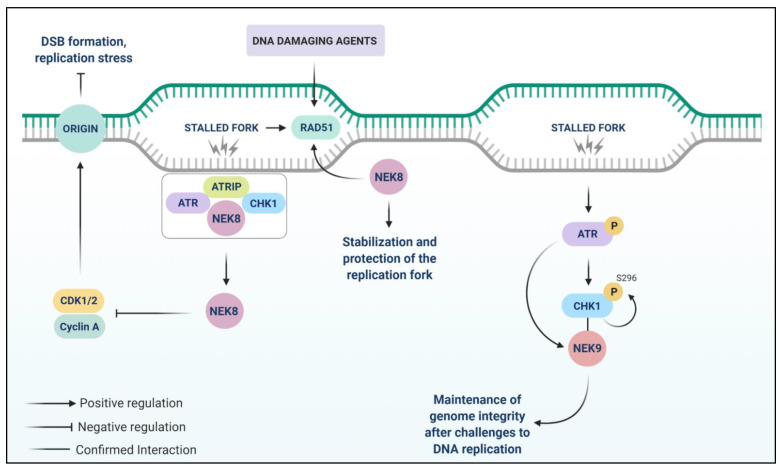
Signaling pathways involving NEK8 and NEK9 in response to replication stress. NEK8 prevents DSB accumulation by suppressing cyclin A-associated CDK activity. NEK8 interacts with ATR, ATRIP (ATR-interacting protein), and CHK1 proteins after DNA damage and is important for the progression of replication forks. NEK8 also regulates RAD51 foci formation upon DNA damage. NEK9 is a replication stress response (RSR) protein since agents involved in replication stress increase its expression. NEK9 interacts with CHK1 and its knockdown decreases CHK1 autophosphorylation (S296) and its kinase activity in response to replicative stress.

**Figure 5 cells-10-00507-f005:**
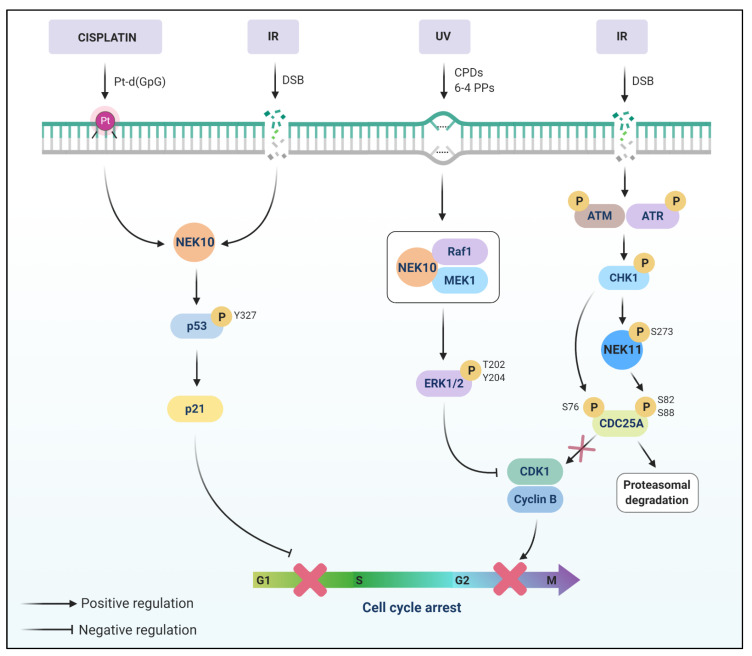
The signaling pathway of NEK10 and NEK11 upon DNA damage. NEK10 assembles in a ternary complex with RAF-1 and MEK1 and mediates ERK1/2 activation, playing a role in the maintenance of the G2/M checkpoint after UV light irradiation. Upon Cisplatin and IR treatments, NEK10 phosphorylates p53 at Y327, increasing p21 levels. p21 is a cyclin-dependent kinase inhibitor that suppresses the cell cycle G1/S phase. NEK10 participates in the control of cell cycle progression through p53 phosphorylation. Upon DNA damage, NEK11 is involved with ATM/ATR checkpoint pathway. Degradation of CDC25A is regulated by NEK11 through the phosphorylation of S82 and S88 of CDC25A, in vivo. CHK1 is responsible for phosphorylating NEK11 at the S273 site in vitro.

## Data Availability

This review does not report any data.

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
