# Peer review of "On Broken Ne(c)ks and Broken DNA: The Role of Human NEKs in the DNA Damage Response"

_cells, 2021, doi:10.3390/cells10030507_

Round 1

Reviewer 1 Report

Here, Pavan et al have compiled the features and functions of 11 NIMA-related kinases (NEKs) in a well structured and formatted manners. This review very well highlighted the diversities of the metabolic functionalities of NEK kinases in the DNA damage response pathways. This study can facilitate the process of drug designing in various aspects of cancer treatment and will be very interesting for the reader of "Cells". I recommend acceptance of this manuscript with a minor English check.

I also want to give a few suggestions which can improve the quality of the manuscript.

The authors should have included the 3-D structures of all the NEK kinases as it will provide a better understanding regarding some discussed facts. Because the authors discussed the structural features.

I also want to see the discussion for the multiple sequence alignments of all the 11 NEKs as it will provide a better picture regarding the evolutionary conservation of this respective class of protein kinases.   

Author Response

Reviewer: 1

The authors should have included the 3-D structures of all the NEK kinases as it will provide a better understanding regarding some discussed facts. Because the authors discussed the structural features.

Answer: We appreciate your opinion on the manuscript. We agree that the addition of structures is useful information for the design of drugs against NEKs. However, we do not focus on the structural aspect, since we have limited information regarding the three-dimensional structures of all NEKs. Only NEK1, NEK2, and NEK7 have a PDB entry and NEK6 has a low-resolution structure. Thus, we have added a table with information on their protein domains and the PDB entry to facilitate research on this field. Finally, given the relevance of NEKs in DDR, we encourage more studies in this area and the discovery of their structures, which is a crucial point for advances in research related to NEKs and diseases. In addition, other recent reviews [1] (e.g. Fry et al., Front Cell Dev Biol, 2017, 5:102) properly addressed the structural aspects of NEKs and we do not wish to repetitive here, but rather focus on the functional aspects in the context of DDR.

I also want to see the discussion for the multiple sequence alignments of all the 11 NEKs as it will provide a better picture regarding the evolutionary conservation of this respective class of protein kinases.

This is another important suggestion but again a little beyond the scope of this review. A phylogenetic study of Neks was previously done [2] (Parker et al., 2007, Plos1, 2(10): e1076), and we now added a couple of phrases to our text where we try to associate the internal family relationship between Neks with possible functional association to specific molecular and DDR/DNA repair-related contexts.

Example of an added phrase that addresses phylogenetic aspects (line 425):

“A previous phylogenetical analysis of all NEKs at large and, specifically, of the vertebrate NEKs had shown the clustering of human Neks in subfamilies (Parker et al., 2007).  Most interestingly, a distinct subfamily cluster is formed by human NEKs 8,9,1,3, and 5. It is remarkable that three of these NEKs: NEK1, 8, and 9, are all somehow functionally involved with the ATR/ATM axis (Fig. 1 and 5)”

Furthermore, we added a new Table 1, where structural aspects and domain organization have been organized in a comparative fashion for all 11 human Nek family members.

References

  1. Fry, A.M.; Bayliss, R.; Roig, J. Mitotic Regulation by NEK Kinase Networks. Front. cell Dev. Biol. 2017, 5, 102, doi:10.3389/fcell.2017.00102.
  2. Parker, J.D.K.; Bradley, B.A.; Mooers, A.O.; Quarmby, L.M. Phylogenetic Analysis of the Neks Reveals Early Diversification of Ciliary-Cell Cycle Kinases. PLoS One 2007, 2, e1076, doi:10.1371/journal.pone.0001076.

Reviewer 2 Report

The review of Pavan et al is the second  review by the same authors about the functions of the NEKs proteins (the first one was published in Molecules Journal in April 2020). The current manuscript  describes a number of very interesting aspect of the NEK family functioning in DNA damage response. However, the way how this review is constructed makes it very difficult to follow. Instead of being a critical view of how NEKs cooperate with each other and with different DDR factors to regulate response to DNA damage, authors provided an endless list of facts, sometimes absolutely unnecessary, which makes the reading very boring and after all somewhat useless. Beyond the mountains of the facts the sense is lost. The review is such form would be accepted as a first draft of a literature review of a PhD student, but not as a consolidate work of twelve researchers working in the field. So, this review absolutely requires a very extensive editing. Just roughly, about 30% of the text can be deleted without losing any essential information. You may start from the abstract – the entire part in the middle, starting from “In summary, we review here that…” till after exposure to DNA damage agents” can be deleted. The readers do not need to read in the abstract the essentials for each of NEK kinase – they have 20 pages of text below! By the way the test is composed of two parts: 1. Introduction – 19 pages; 2. Conclusion – 12 lanes. Some information can be easily organized in the form of Tables or Figures. For example, if authors consider that it is absolutely necessary for the readers to know the molecular weight, number of amino acids and localization of each NEK, they can put this information in a Table. The same, the secondary structure of NEKS with all the domain annotated can be shown in a Figure. All that will be way more informative, than the paragraphs of text at the beginning of every NEK sub-chapter. May be another Figure demonstrating the principles of DNA damage response will be useful, which can help to avoid massive repetitions about DDR mechanisms all through the text.

Some of the definitions and abbreviations are given multiple times (e.g., “DNA damage response (DDR) – in the abstract, at the page 2, in the legend of the Figure 1 and Figure 3; ATM and ATR definitions (page 2, page 5);  the definition/abbreviation for “Ionizing Radiation (IR)” is mention twice in the page 2 and each time in the Figures 1, 2, 3, 5 ). By the way, because many abbreviations are already introduced in the text, it is not necessary to show them in the Figures, which overloads images with information. These are the several examples and there are many more, so authors have to go through the text and eliminate all these unnecessary repetitions.

The Figures, generally quite charged, can also be made easier to follow. For example, only one “Ionizing Radiation (IR)” annotation can be left on the Figure 1. A simple arrow from DSB site to ATM would be enough. The lines delineating “confirmed interaction” and “hypothesis” are not present in every figure, so the figures have to be checked and unnecessary annotations deleted. “Figure created using BioRender” can be mention only once in Acknowledgements section, for example, and not at the end of each Figure. Also, do not repeat at the end of each sub-chapter the same leitmotif: “taken together these data show importance of (Nek9 or NEK10 or NEK 11 etc.) in DDR”.

To summarize, this review requires a major revision: do a very extensive editing work, at least 30-40% less of the text, give the manuscript for the corrections to native English speakers (there are many gramma mistakes), tighten Figures, introduce summarizing Tables and Figures.

Author Response

Reviewer: 2

“So, this review absolutely requires a very extensive editing. Just roughly, about 30% of the text can be deleted without losing any essential information. You may start from the abstract – the entire part in the middle, starting from “In summary, we review here that…” till after exposure to DNA damage agents” can be deleted. The readers do not need to read in the abstract the essentials for each of NEK kinase.”

Answer: We agree with that point. The abstract has been reorganized to make it shorter. Also, we reorganized non-direct data related to DDR in a table, which allowed the reduction of several lines of the manuscript.

“By the way the test is composed of two parts: 1. Introduction – 19 pages; 2. Conclusion – 12 lanes. Some information can be easily organized in the form of Tables or Figures. For example, if authors consider that it is absolutely necessary for the readers to know the molecular weight, number of amino acids and localization of each NEK, they can put this information in a Table. The same, the secondary structure of NEKS with all the domain annotated can be shown in a Figure. All that will be way more informative, than the paragraphs of text at the beginning of every NEK sub-chapter.

Answer: The characterization of NEK proteins is indeed important since NEKs are not well studied compared to other kinases, and a greater focus needs to be given regarding their basic information. As suggested, we added a table with the following information: Gene location (chromosome), amino acids, molecular weight, functions, subcellular location, protein domains, 3D structure, method of obtaining the 3D structure, and PDB entry. This information was removed from the text.

“May be another Figure demonstrating the principles of DNA damage response will be useful, which can help to avoid massive repetitions about DDR mechanisms all through the text. Some of the definitions and abbreviations are given multiple times (e.g., “DNA damage response (DDR) – in the abstract, at the page 2, in the legend of the Figure 1 and Figure 3; ATM and ATR definitions (page 2, page 5);  the definition/abbreviation for “Ionizing Radiation (IR)” is mention twice in the page 2 and each time in the Figures 1, 2, 3, 5 ). By the way, because many abbreviations are already introduced in the text, it is not necessary to show them in the Figures, which overloads images with information. These are the several examples and there are many more, so authors have to go through the text and eliminate all these unnecessary repetitions.”

Answer: We re-evaluated the manuscript and figures and corrected all unnecessary repetitions and redundant information. We believe the text and figures have now become cleaner. We deleted redundant portions of the figure legends.

“The Figures, generally quite charged, can also be made easier to follow. For example, only one “Ionizing Radiation (IR)” annotation can be left on the Figure 1. A simple arrow from DSB site to ATM would be enough. The lines delineating “confirmed interaction” and “hypothesis” are not present in every figure, so the figures have to be checked and unnecessary annotations deleted. “Figure created using BioRender” can be mention only once in Acknowledgements section, for example, and not at the end of each Figure. Also, do not repeat at the end of each sub-chapter the same leitmotif: “taken together these data show importance of (Nek9 or NEK10 or NEK 11 etc.) in DDR”.”

Answer: We agreed that the figures could be improved and made the suggested changes to make them less charged. We have also removed repeated sentences like references from the Biorender program and phrases that do not add informative content.

“To summarize, this review requires a major revision: do a very extensive editing work, at least 30-40% less of the text, give the manuscript for the corrections to native English speakers (there are many gramma mistakes), tighten Figures, introduce summarizing Tables and Figures.”

Answer: Grammatical and concordance errors were corrected by an English native professional (see attached letter , here) and the certificate of this correction is attached to the resubmission of this manuscript. Several lines of text have been removed given the introduction of a table, which summarized basic data of NEKs. In total, we deleted roughly 20 % of the text that was either redundant or less relevant. The figures were adjusted to converge common points of the signaling pathways. We did not insert an additional figure representing the domains of NEKs because this information was added in the form of a table and is presented in other reviews.

Reviewer 3 Report

Pavan et al summarized the roles of NEK kinases in DNA damage response and DNA repair. The topic was interesting and the authors provided nice illustration of NEK related DNA damage signaling pathway. However, the review is sloppy and poorly written, which includes dozens of grammatical mistakes. Furthermore, the authors listed all relevant publications, but did not organized them logically. The authors need professional language assistance to improve the manuscript.

Author Response

“Pavan et al summarized the roles of NEK kinases in DNA damage response and DNA repair. The topic was interesting and the authors provided nice illustration of NEK related DNA damage signaling pathway. However, the review is sloppy and poorly written, which includes dozens of grammatical mistakes. Furthermore, the authors listed all relevant publications, but did not organized them logically. The authors need professional language assistance to improve the manuscript.”

Answer: We appreciate your opinion on the manuscript and agree with the comments. Thus, we forwarded the manuscript to an English native professional service (see attached letter), which performed deep corrections of grammatical and concordance errors. We also improved the text and now believe that the reading is more fluid and the content more connected. We also deleted 20% of the text deemed as redundant or superfluous.

Round 2

Reviewer 2 Report

The authors have addressed all my comments and now the review can be published.